# When Pediatric Headaches Are Not Benign—Eye Findings

**DOI:** 10.3390/children10020372

**Published:** 2023-02-14

**Authors:** Sam Karimaghaei, Brita S. Rook

**Affiliations:** 1Department of Ophthalmology, Harvey and Bernice Jones Eye Institute, University of Arkansas for Medical Sciences, Little Rock, AR 72205, USA; 2Department of Ophthalmology, Arkansas Children’s Hospital, Little Rock, AR 72202, USA

**Keywords:** headache, papilledema, infection, autoimmune, stroke, arteriovenous malformation, hydrocephalus, neoplasia, idiopathic intracranial hypertension

## Abstract

Headache is the most common neurologic complaint that presents to the pediatrician. While most headaches are benign in nature, patients must be carefully evaluated to rule out life- or vision-threatening causes. Non-benign etiologies of headache may exhibit ophthalmologic signs and symptoms that can help narrow the differential diagnosis. It is also important for physicians to know in what situations appropriate ophthalmologic evaluation is necessary, such as evaluating for papilledema in the setting of elevated intracranial pressure. In this article we discuss life- and/or vision-threatening etiologies of headache, including infection, autoimmune disease, cerebrovascular pathologies, hydrocephalus, intracranial neoplasia, and idiopathic intracranial hypertension, and their associated ophthalmologic manifestations. Due to less familiarity of the disease amongst primary care providers, we discuss pediatric idiopathic intracranial hypertension in more comprehensive detail.

## 1. Introduction

The most common neurologic symptom in childhood is headache. It has an estimated prevalence of 58.4% in the pediatric population and is a leading cause of clinic visits and pediatric emergency department admissions [1,2]. While most pediatric headaches are benign in nature, either related to a primary headache disorder (i.e., migraine, tension-type headache) or a self-limited viral illness, some have much more sinister etiologies [1]. It is important for the pediatrician to be aware of these non-benign headaches, and initial work-up should focus on ruling out life- or vision-threatening diagnoses. 

Vision- or life-threatening causes of headache may present with ophthalmic findings that can alert the pediatrician to their likely presence. Therefore, it is of high value to discuss the various etiologies of non-benign headaches and associated ocular findings. 

Non-benign headaches can be secondary to multiple different etiologies, including infectious, autoimmune, cerebrovascular, structural, and neoplastic. Among adults, idiopathic intracranial hypertension, or pseudotumor cerebri, is another known vision-threatening cause of headache, typically affecting overweight or obese women of reproductive age. However, it can also occur in children of all ages and is important to keep in the differential for any pediatric patient presenting with headache [3].

In this article, we discuss non-benign causes of secondary headaches and associated ocular findings, with additional focus on pediatric idiopathic intracranial hypertension. 

## 2. Intracranial Hypertension and Papilledema

Many non-benign headaches involve a rise in intracranial pressure (ICP). Elevated ICP is associated with a constellation of hallmark signs and symptoms, which may include headaches generally worsened by lying down and Valsalva maneuvers, nausea, vomiting, and pulsatile tinnitus. Ophthalmic manifestations of high ICP include cranial nerve six palsy, which may present with diplopia, and papilledema [4].

Papilledema is specifically defined as optic disc swelling secondary to raised ICP and may lead to vision loss, even in cases of only mild ICP elevation. Acutely, papilledema may manifest as transient visual obscurations (TVOs) [5]. TVOs may be partial or complete and last seconds to minutes [5,6]. TVOs occur in about 68% of patients with papilledema, yet the pathophysiology remains unclear. Hypothesized mechanisms include transient ischemia or myelin distortion at the nodes of Ranvier. Papilledema may infrequently cause acute disc ischemia, which most often causes inferonasal visual field defects but may occasionally lead to sudden loss of central vision. Chronic papilledema may cause progressive loss of retinal nerve fibers entering the optic disc. Loss of peripheral nerve fibers occurs earlier and to a greater extent than central fibers. Visual field defects are usually arcuate-shaped and generally involve the inferior peripheral vision due to preferential loss of superior nerve fibers [5]. Visual acuity is usually preserved in early to moderate papilledema. Only in severe cases or if exudates impair retinal function does visual acuity deteriorate significantly [4]. 

Ophthalmologists generally use slit lamp and indirect ophthalmoscopy to evaluate the optic nerve. While primary and emergency care providers are usually not trained in these advanced fundoscopic techniques or do not have access to the necessary equipment, direct ophthalmoscopes are readily available in most healthcare settings. Direct ophthalmoscopy is intended for examination of the optic disc and should always be performed if elevated ICP is suspected to evaluate for presence of optic disc edema [4]. Clinically detectable papilledema usually develops within one to seven days of ICP elevation but does typically lag behind the elevation in ICP [5]. It is usually bilateral but may also be unilateral in some cases [4].

The direct ophthalmoscope produces a 15× magnified, monocular, upright image of the retina. A normal optic disc appears orange-pink with well-defined sharp borders and a pale central cup [4]. Optic disc swelling in papilledematous eyes can range from mild to severe. Swelling usually begins at the lower pole of the optic disc and progresses over the upper pole to the nasal pole and may eventually cause temporal disc elevation. Frank blurring of the optic disc margins occurs after significant disc edema accumulates [5]. Additionally, retinal vessels entering and exiting the disc may be dilated, tortuous, or obscured by edematous tissue. Hemorrhages of the retinal nerve fiber layer or cotton wool spots may occur in severe cases of papilledema [4].

Ultimately, when a patient’s symptoms suggest intracranial hypertension, optic disc edema on fundoscopy is a strong predictor of elevated ICP and warrants urgent neuroimaging and possible lumbar puncture. It should be noted, however, that absence of optic disc edema does not rule out intracranial hypertension. Compensatory mechanisms may prevent development of papilledema if the rise in ICP is gradual or moderate, and papilledema rarely occurs in eyes with preexisting optic atrophy. In patients with previously documented normal optic discs, a new appearance of disc swelling in the context of concerning clinical history has the highest positive predictive value for ICP elevation [4].

## 3. Non-Benign Etiologies of Headache

### 3.1. Infection

CNS infections presenting as cerebral abscess, meningitis, or encephalitis require prompt diagnosis and treatment. 

#### 3.1.1. Brain Abscess

A life-threatening infection in children, brain abscesses result from seeding of the cerebral parenchyma with microorganisms via contiguous spread from nearby infection (i.e., mastoiditis, chronic otitis media, sinusitis, and meningitis), hematogenous spread from distant sites in the setting of cyanotic heart disease, penetrating head trauma, neurosurgical procedures, and cryptogenic sources. Clinical findings are generally related to the effect of a space-occupying mass, focal neuronal dysfunction of involved parenchyma, and/or signs and symptoms of the predisposing infection. Headache is the most common initial symptom in older children and adolescents, while infants and small children more commonly display irritability. Absence of fever does not rule out brain abscess as the cause of headache, as it may be absent in up to half of affected children. Possible ocular manifestations are related to both elevation of intracranial pressure (ICP) and region of parenchymal involvement. Papilledema is present in less than 25% of cases. However, its presence on exam requires immediate computed tomography (CT) imaging and neurosurgical evaluation given the risk for herniation [7,8]. Abscesses involving the temporal lobe can cause visual field defects ranging from a contralateral homonymous superior quadrantanopia to complete homonymous hemianopia. Coinciding dysphasia may be present if the dominant hemisphere is involved. Parietal lobe involvement may present with visual field defects ranging from contralateral inferior quadrantanopia to complete homonymous hemianopia. Dominant hemisphere involvement may also coincide with dysphasia, while non-dominant hemisphere involvement may present with dyspraxia and spatial neglect [8]. In addition to gait ataxia, nystagmus and defective conjugate eye movements are indicative of cerebellar abscess [7,8]. Brainstem abscesses often present with a combination of cranial nerve palsies and deficits of the ascending sensory and descending motor pathways [8]. 

#### 3.1.2. Meningitis

In infants and children, meningitis usually occurs when encapsulated bacteria colonize the nasopharynx and hematogenously spread to the meninges. However, it may also occur from contiguous spread of infection to the meninges from the paranasal sinuses or middle ear via the mastoid process, severe head trauma with skull fracture, cerebrospinal fluid (CSF) rhinorrhea, as well as by direct bacterial inoculation of CSF by congenital dural defects, neurosurgical procedures, and penetrating wounds [9]. 

Papilledema may induce visual symptoms in patients presenting with meningitis. Poor CSF absorption due to inflammation at the level of the arachnoid granulations, obstructive hydrocephalus, and secondary cerebral edema may all contribute to ICP elevation in the setting of meningitis. However, the reported incidence of papilledema with meningitis is relatively low, and cases of meningitic papilledema are generally mild and transient [10]. In a study by Hanna et al., only 2.5% of 2178 cases of meningitis had concurrent papilledema. However, among patients with tuberculous meningitis, up to 25% had papilledema [11]. It is important to note, however, that optic disc swelling in patients with meningitis may not necessarily be solely attributable to high ICP, as spread of infection or a secondary inflammatory process may induce inflammation of the optic discs [10].

While a less common cause of meningitis in developed countries but prevalent in endemic regions, tuberculous meningitis is a serious infection notorious for devastating visual complications. While visual pathway involvement is a well-established association, it is not always given sufficient attention in clinical practice and may go unnoticed in patients presenting with this disease, especially those with altered mentation. Up to 14% of tuberculous meningitis survivors have some degree of permanent, disabling vision loss. The cause of visual complications is not exactly known, but several mechanisms have been implicated, including strangulation of the optic nerves and chiasm by thick exudates, papillitis and retrobulbar optic neuritis due to mycobacterial invasion of the optic nerve, optic chiasm compression from above by dilated third ventricle in the setting of hydrocephalus, papilledema, optic nerve and chiasm infarction secondary to tuberculous endarteritis of vasa nervosa, optic nerve tuberculoma, and tuberculous uveitis. In general, any structure of the visual pathway can be affected, but the optic nerve and chiasm are the most frequently involved. While rare, optic radiation and visual cortex involvement present as cortical blindness, which is defined as vision loss with sparing of pupillary light responses. Causes of optic radiation or visual cortex involvement may include basilar artery involvement, enlarging hydrocephalus, or occipital lobe tuberculoma [12].

#### 3.1.3. Encephalitis

Infectious encephalitis is another life-threatening cause of headache in children and involves ICP elevation. The most common culprit pathogen is herpes simplex virus (HSV). Children classically present with a brief flu-like prodrome followed by severe headaches, nausea, vomiting, and altered level of consciousness. Patients may also experience focal neurologic deficits and seizures. Many features may overlap with those of meningitis, including fever, headache, seizures, and meningismus. A minority of patients, such as immunocompromised children, may present more subacutely. They may be afebrile or have a low-grade fever and develop behavioral or speech disturbances that progress to more frank encephalopathy or seizure [13]. Given that cerebral swelling may elevate ICP, papilledema is a potential complication that should be evaluated. The incidence varies in the literature, but Pedersen found that 21% of 80 patients with presumed aseptic encephalitis had papilledema [14].

### 3.2. Autoimmune Disease

In addition to infectious etiologies, autoimmune inflammatory causes should also be considered in patients suspected to have encephalitis. One such clinical entity is myelin oligodendrocyte glycoprotein (MOG) antibody (Ab)-positive autoimmune encephalitis (AE). Children may present with prodromal symptoms frequently associated with severe headache, encephalopathy, somnolence, focal neurologic symptoms, and seizures [15]. Anti-N-methyl-D-aspartate receptor (NMDAR) encephalitis also presents with prodromal symptoms that resemble viral infection (i.e., fever, headache, and malaise), followed by the onset of neuropsychiatric symptoms (i.e., visual or auditory hallucinations, bizarre behavior, anxiety, aggressiveness, delusional or paranoid thoughts, etc.), insomnia, speech disturbances, focal neurologic symptoms, dyskinesias, memory issues, seizures, altered level of consciousness, autonomic instability, and central hypoventilation [16,17]. Autonomic instability and hypoventilation tend to be less severe in children compared to adults. Among adult women, anti-NMDAR encephalitis frequently occurs in the setting of an underlying NMDAR-positive tumor, usually ovarian teratomas. Its association with tumor is rare among men and children [16]. Acute disseminated encephalomyelitis (ADEM) is a demyelinating CNS disorder that can present in children, most often following symptoms of a systemic viral illness (55–85% of pediatric cases). Less frequently it is associated with preceding vaccination (4–18% of pediatric cases). However, the presence of a true causal relationship between vaccination and pediatric ADEM is unclear. There is considerable overlap between MOG Ab-positive AE and ADEM. In fact, MOG IgG-associated disease is the underlying cause for many cases of ADEM in children. ADEM often presents with a prodrome of fever, headache, and nausea that begins, on average, 12 days after an inciting viral illness and lasts for about 3–4 days. The prodromal period is followed by onset of neurologic symptoms, most commonly encephalopathy, pyramidal dysfunction, cerebellar signs, and cranial nerve deficits. However, the range of neurologic manifestations is broad [18]. A possible ophthalmic manifestation of pediatric ADEM is optic neuritis, which occurs in 1–15% of cases [18,19]. The hallmark features of optic neuritis are decreased visual acuity, abnormal color vision (most frequently red color desaturation), visual field defects, and painful eye movements. Pertinent physical exam findings of acute optic neuritis include relative afferent pupillary defect (RAPD) and blurring of the optic disc margins. However, if optic neuritis is bilateral, as can be the case in ADEM, there may be no RAPD [19]. Up to 25% of children with ADEM require pediatric intensive care unit (PICU) admission due to severe encephalopathy, seizures, and diaphragmatic paralysis, and 75% of these patients require mechanical ventilation. Fortunately, however, an acute ADEM episode appears to only rarely result in death [18]. 

### 3.3. Cerebrovascular Diseases

Cerebrovascular pathology must be considered in pediatric patients presenting with headache. Pediatric strokes are relatively rare but associated with considerable morbidity and mortality. About 10–25% of pediatric patients with stroke die, up to 25% have a recurrence, and up to 66% acquire persistent neurologic deficits or a post-stroke seizure, learning, or developmental disorder. Acute ischemic stroke (AIS) accounts for about 50% of strokes in the pediatric population, in contrast to 80–85% of strokes in adults [20]. 

In children, AIS most often presents with focal neurologic deficits, the most common being hemiplegia. Hemorrhagic strokes usually present as headaches or altered levels of consciousness. However, ischemic strokes can present with headache as well. Seizure occurs in about half of children with stroke and is common in both ischemic and hemorrhagic types [20].

Clinical presentation of stroke varies significantly among pediatric age groups, and signs and symptoms are more nonspecific the younger the child is. This often leads to a delay in diagnosis that can be mitigated by understanding common risk factors and etiologies, which differ significantly from adults. While hypertension, diabetes, and atherosclerosis account for the majority of adult strokes, childhood strokes generally have distinctly different etiologies. 

Cardiac etiologies, which account for up to one-third of childhood ischemic strokes, include cyanotic heart disease, patent foramen ovale, cardiomyopathies, rheumatic heart disease, prosthetic valves, and endocarditis. Sickle cell disease and prothrombotic disorders, such as Factor V Leiden and Protein C and S deficiencies, are common hematologic causes of pediatric ischemic stroke. Clotting factor deficiencies, such as Factor VII and VIII deficiencies, can lead to hemorrhagic strokes. 

A wide range of infections can lead to stroke, including HIV, mycoplasma, chlamydia, enterovirus, parvovirus B19, influenza A, coxsackie virus, Rocky Mountain spotted, and cat-scratch disease [20]. Another important cause to be aware of is varicella zoster (VZV) vasculopathy. Studies have shown an increased risk of stroke in patients after herpes zoster infection, with the greatest incidence occurring within the first 2 weeks after infection and in patients with herpes zoster ophthalmicus [21]. A range of 5–12% of pediatric cases of bacterial meningitis and viral encephalitis will have a complication of stroke due to development of local vasculitis and thrombosis [20].

Arteriovenous malformations (AVM) are the most common cause of hemorrhagic strokes in children after infancy but can also cause ischemic strokes. AVM may occur in the context of rare neurocutaneous disorders, such Sturge–Weber syndrome, hereditary hemorrhagic telangiectasia, von Hippel Lindau syndrome, and neurofibromatosis [20]. Moyamoya is another AVM seen in children and can be life- or vision-threatening [22,23]. It is characterized by progressive stenosis of the intracranial internal carotid arteries and their proximal branches, resulting in compensatory development of small vessel collaterals near the carotid apex, cerebral cortex, leptomeninges, and external carotid artery branches supplying the dura and skull base. The angiographic appearance of the network of dilated collateral vessels is akin to “a puff of cigarette smoke”, or *moyamoya* in Japanese. While moyamoya is seen in greater prevalence in people of Asian heritage and is the most common pediatric cerebrovascular disease in Japan, it has been observed in people of various ethnic backgrounds across the world. Peak incidence occurs in children around 5 years old and adults in their mid-40s. Emergent presentations include symptoms of cerebral ischemia from progressive arterial stenosis or intracranial hemorrhage (i.e., subarachnoid, intraparenchymal, intraventricular) from fragile collaterals. A common ambulatory presentation is a typically migraine-like headache from dilated transdural collateral vessels. In children, transient ischemic attacks (TIA) or ischemic stroke may be induced by dehydration, exertion, induction of anesthesia, or even hyperventilation with crying [22]. Moyamoya disease is rarely associated with certain ophthalmic findings. The most common is morning glory disc anomaly (MGDA), a rare congenital malformation of the optic nerve characterized by an enlarged optic disc with peripapillary hyperpigmentation, retinal vessels radiating from the periphery rather than center of the disc, and a central glial tuft. The disc anomaly is named for its resemblance of the morning glory flower [24]. Identification of MGDA on fundoscopic exam warrants cerebrovascular imaging to rule out concomitant moyamoya [22]. Other reported ophthalmic findings include retinal vascular occlusion, particularly central retinal artery occlusion, ocular symptoms secondary to the ischemia or hemorrhage typical of moyamoya vascular changes (i.e., amaurosis fugax, visual field defects, cortical blindness, etc.), optic disc pallor, congenital cataract, and others [23].

Genetic syndromes and metabolic disorders may lead to stroke in children as well. Homocysteinuria predisposes children to AIS and is often associated with both lens dislocation and intellectual disability. Marfan syndrome is also associated with ectopia lentis and may rarely cause ischemic stroke. Children with tuberous sclerosis have a higher risk of embolic and hemorrhagic strokes. Cerebral vasculitis, either idiopathic or in the context of rheumatologic diseases, is an uncommon cause of pediatric stroke and is generally more common in children older than 14 years [20].

### 3.4. Hydrocephalus

Hydrocephalus is defined as ventricular distention resulting from mismatch between CSF production and CSF absorption or compromise in CSF flow through the ventricular system [25]. It can have various etiologies, both congenital and acquired [26]. About 55% of cases are congenital [25]. Acquired causes may be secondary to intraventricular hemorrhage, trauma, tumors, or infection [26]. In children up to 2 years old with open fontanelles, characteristic signs of hydrocephalus include head enlargement, wide anterior fontanelle, prominent scalp veins, increased muscle tone, irritability, and vomiting. Ophthalmologic signs include setting-sun eyes, nystagmus, and optic nerve atrophy [25,27]. The setting-sun eye phenomenon describes a persistent downward gaze due to upward gaze paresis. The sclera between the iris and upper eyelid may be seen, and the lower eyelid may cover part of the pupil. It is thought to be caused by compression of periaqueductal structures from aqueductal distention secondary to intracranial hypertension. Persistent setting-sun eyes is one of the most common signs of ICP elevation in pediatric patients, occurring in 40% of children with hydrocephalus and 13% with ventriculoperitoneal shunt failure. In fact, it is one of the earliest signs of hydrocephalus, usually appearing before detectable head circumference enlargement, full fontanelle, separation of sutures, irritability, or vomiting. Therefore, identification of persistent setting-sun eyes in a child requires urgent neuroimaging and neurosurgical evaluation [27]. Children older than 2 years may also present with the above symptoms. However, because most will have complete closure of fontanelles by this age, increasing head circumference generally does not occur [25]. Typical manifestations of hydrocephalus after infancy include some combination of headache, vomiting, diplopia, and papilledema. Diplopia is usually secondary to cranial nerve VI palsy [28]. Optic atrophy, hypothalamic dysfunction and spastic lower limbs may also be seen in children over 2 years old [25]. Papilledema is frequently absent in the setting of hydrocephalus. In a study of 46 children with hydrocephalus, Lee et al. found that papilledema was more common among children who were older, had higher ICP, and whose hydrocephalus was due to a brain tumor. Papilledema was absent in 41% of children [29].

### 3.5. Neoplasia

The most common solid tumors in children are CNS tumors, and brain tumors are the leading cause of death among all childhood cancers. The most common sites for brain tumors are supratentorial in children up to 3 years of age and after 10 years of age, and infratentorial in children between 4 and 10 years of age [30]. Signs and symptoms depend on patient age and location of the tumor. Infants generally present with nonspecific symptoms, such as irritability, listlessness, failure to thrive, loss of developmental milestones, macrocephaly, and vomiting [30,31]. Young children are more likely than infants to present with localizing neurologic symptoms, but many manifest typical findings of elevated ICP, including progressively worsening headaches, nausea, and vomiting, in the absence of focal neurologic deficits. Older children are the most likely to manifest localizing neurologic findings. Both supratentorial and infratentorial tumors often cause intracranial hypertension by inducing an obstructive hydrocephalus. Infratentorial tumors generally cause obstruction at the level of the fourth ventricle [31].

#### 3.5.1. Supratentorial Tumors 

Supratentorial tumors produce neurologic deficits depending on the surrounding structures they compress or infiltrate. Masses around the optic chiasm and hypothalamus (i.e., craniopharyngiomas, chiasmatic–hypothalamic gliomas) often cause visual field deficits (i.e., bitemporal hemianopia), disruption of the hypothalamic–pituitary axis, and/or changes in appetite or behavior [31]. Neoplastic compression of the dorsal midbrain, often by pineal gland tumors, may cause Parinaud’s syndrome, or dorsal midbrain syndrome [31,32]. It is characterized by a unique constellation of neuro-ophthalmic findings, including upward gaze palsy (setting-sun eyes), bilateral upper eyelid retraction (Collier’s sign), light near dissociation, and convergence retraction nystagmus. Light near dissociation is noted when the pupils constrict normally with accommodation, but there is impairment of the pupillary light reflex [32]. Convergence retraction nystagmus (CRN) is specific to midbrain lesions, but is not a true nystagmus as the name implies. It is a co-contraction of the medial and lateral rectus muscles that manifests as convergence and globe retraction that is more pronounced in upward gaze [33,34]. 

#### 3.5.2. Infratentorial Tumors

Clinical manifestations of infratentorial tumors can vary considerably between different types. Diffuse intrinsic brainstem gliomas usually cause rapidly progressive cranial neuropathies in conjunction with pyramidal and/or sensory long tract involvement [31]. Symptoms of cerebellar astrocytomas may include ataxia, nystagmus, intracranial hypertension from fourth ventricle compression, and neck pain from tonsillar herniation [31,35]. Medulloblastomas present similarly to cerebellar astrocytomas, but are more rapidly progressive. Ependymomas usually arise within the floor of the fourth ventricle and often first present with nausea and vomiting due to compression or invasion of the area postrema. They eventually cause occlusion of the fourth ventricle as they continue to grow [31].

#### 3.5.3. Pituitary Apoplexy

Pituitary apoplexy describes the rapid onset of signs and symptoms secondary to hemorrhage or infarction of a pituitary tumor. It is a rare occurrence in children, but presenting symptoms to be aware of include rapid onset headache, nausea, vomiting, dizziness, altered mental status, coma, and pituitary insufficiency. Possible ophthalmic findings are visual field deficits, including bitemporal hemianopia, ophthalmoplegia, and photophobia [36].

### 3.6. Pediatric Idiopathic Intracranial Hypertension

Another important consideration for pediatric patients presenting with headache is idiopathic intracranial hypertension (IIH), also known as pseudotumor cerebri. Because the disease is predominantly seen in overweight or obese women of childbearing age, most physicians are perhaps less familiar with its occurrence in children. As such, we will discuss this entity in greater detail.

#### 3.6.1. Disease Entity

IIH is a clinical syndrome defined by elevated intracranial pressure in the absence of identifiable pathology on neuroimaging or CSF analysis, including tumors, hydrocephalus, structural or vascular lesions, or infection [3]. It is a diagnosis of exclusion and, therefore, requires all etiologies of headache described previously be ruled out. 

#### 3.6.2. Demographics

The characteristic demographics of children with IIH depend on whether they are pre- or post-pubertal. Like adults, post-pubertal children with IIH are typically obese and female. However, pre-pubertal IIH is less frequently associated with obesity and occurs in males and females with equal frequency [37].

#### 3.6.3. Pathophysiology

The exact cause of IIH is unknown and a number of hypotheses exist, including decreased CSF absorption at the level of arachnoid villi. Another proposed mechanism is elevated intracranial venous pressure reducing the pressure gradient for CSF flow across the arachnoid villi. Identification of cerebral venous sinus stenosis in some patients has supported this theory, but evidence that the stenosis may resolve with normalization of ICP suggests that sinus narrowing may actually be a result of rather than cause of IIH in some cases [3].

While no obvious etiology is identified in many cases of IIH, certain medications and diseases have been associated with IIH in both adults and children. Medications and conditions commonly implicated include tetracycline antibiotics, vitamin A, retinoids, growth hormone, thyroid hormone replacement, withdrawal from chronic steroids, primary adrenal insufficiency, and anemia [37].

#### 3.6.4. Clinical Presentation

IIH causes signs and symptoms of elevated ICP. The most common symptom among children is headache and occurs in around 90% of cases [3]. Headaches are generally intermittent, diffuse, throbbing, and worse upon awakening. Patients may also complain of retro-orbital, neck, or back pain [38]. Nausea, vomiting, tinnitus, fatigue, neck stiffness, decreased appetite, vertigo, somnolence, limb paresthesias, ataxia, and irritability are other possible systemic symptoms [3]. Young children may only present with irritability. Children with IIH typically have a normal level of consciousness and absence of seizures or focal neurologic deficits. The presence of these symptoms should increase suspicion for an intracranial tumor [38]. Characteristic ophthalmologic symptoms include transient visual obscurations, vision loss (blurry vision, decreased visual acuity, and visual field deficits), photophobia, and diplopia [3,38]. In fact, vision loss may be the only presenting symptom in some children. Therefore, absence of headache should not be used to exclude IIH [38]. Papilledema is the most common and most concerning clinical sign of IIH in children, as it can lead to permanent vision loss if left untreated. It is generally bilateral, but its severity can be asymmetric. Only rarely is papilledema unilateral. It is important to be aware that papilledema in the absence of any symptoms may be the sole manifestation of IIH in children. Therefore, identification of optic disc edema in a completely asymptomatic child still warrants prompt investigation [3].

IIH also commonly presents with sixth nerve palsy and in higher frequency among children than adults. In fact, cranial nerve palsy occurs in up to 48% of pediatric patients with IIH. Sixth nerve involvement occurs due to its long intracranial course, and the palsy can be unilateral or bilateral. Clinically, it manifests as esotropia and/or diplopia. Normalization of ICP will improve cranial nerve dysfunction, so in addition to its therapeutic effect, it is diagnostically useful to exclude alternative etiologies of cranial nerve deficits [3].

#### 3.6.5. Diagnostic Testing

Patients suspected of having IIH require magnetic resonance imaging (MRI) of the brain and orbits with and without gadolinium contrast to rule out ICP raising intracranial lesions, including tumors, hydrocephalus, and structural or vascular anomalies, and magnetic resonance venography (MRV) to rule out cerebral venous sinus thrombosis. If neuroimaging is negative for obvious intracranial pathology, lumbar puncture should follow to rule out meningitis and confirm elevated CSF pressure [3]. For children aged 1 to 18 years, a CSF opening pressure above 28 cm H_2_O is considered elevated. In neonates, an opening pressure above 7.6 cm H_2_O is considered elevated. It should also be noted that when evaluating for meningitis, normal CSF reference ranges are different in neonates and young infants as compared to older children and adults [39]. While neuroimaging in IIH, by definition, should be normal, there are subtle signs of intracranial hypertension that can be identified on MRI. Signs of elevated ICP include posterior flattening of the globes, an empty sella turcica, distention of the perioptic subarachnoid space, enhancement of the prelaminar optic nerve, vertical tortuosity of the orbital optic nerve, venous sinus stenosis, and intraocular protrusion of the optic nerve [3].

#### 3.6.6. Treatment 

Treatment of IIH focuses on preserving vision, the most worrisome consequence of IIH, and managing headaches. Patients should be managed by a neuro-ophthalmologist or co-managed by a neurologist and ophthalmologist. Patients who do not present with significant or rapidly progressive visual decline are initially managed conservatively with medication. Observation alone can be considered, but only in mild cases without impending vision loss. Obese or overweight children should attempt to lose weight. Some neuro-ophthalmologists recommend 10% weight reduction from that measured at the time of diagnosis [39]. Even 3–6% weight loss has been shown to significantly improve papilledema. Additionally, potential causative factors of IIH should be identified and addressed, including discontinuation of associated medications whenever possible [3]. 

First line medical therapy is the carbonic anhydrase inhibitor acetazolamide. By lowering ICP through reduction in CSF production, it improves headache, reduces papilledema, and stabilizes vision. Common adverse effects of acetazolamide to monitor include electrolyte abnormalities, anorexia, and paresthesias of the extremities. Furosemide can be used as a substitute or adjunctive therapy when acetazolamide alone fails to improve symptoms or causes intolerable side effects. Topiramate also reduces ICP and can be used to treat headaches. It can also cause weight loss through appetite suppression. Chronic steroids are not indicated for long-term management of IIH due to their side effect profile, including weight gain. However, short-term use of IV methylprednisolone with acetazolamide can be considered to stabilize vision of children with more severe papilledema while they wait for surgery. Non-steroidal anti-inflammatory drugs may reduce headaches in some patients [3].

Surgical management of IIH should be considered when symptoms are refractory to medical therapy or the risk for permanent vision loss is high (i.e., rapidly progressive vision loss, significant visual impairment on presentation). Surgical options include optic nerve sheath fenestration (ONSF) and CSF shunting. ONSF is favored over CSF shunting when vision loss is the primary manifestation of IIH, as it is effective in most patients and carries a lower complication rate. CSF shunting is performed for intractable headaches refractory to medical therapy or when ONSF fails to resolve papilledema or stabilize vision [39].

#### 3.6.7. Prognosis

Pediatric IIH generally carries a favorable prognosis. However, permanent vision loss still occurs in a fraction of patients who receive appropriate treatment. Less than 10% experience permanent loss of visual acuity and around 17% have persistent visual field deficits [39]. In contrast to the chronic course typically seen in adults, IIH in children tends to resolve more quickly with appropriate management [3]. Papilledema, on average, resolves in 4.7 months, and the risk of disease recurrence is less than 22%. Children who improve with weight loss should be counseled that regaining lost weight might result in disease relapse [39].

## 4. Conclusions

Headache is one of the most common chief complaints seen by the pediatrician. While most are benign, some are manifestations of serious disease processes that can be life- or vision-threatening. Etiologies of non-benign headaches in children can be categorized into several groups—CNS infection, autoimmune disease, cerebrovascular pathology, hydrocephalus, intracranial tumors, and idiopathic intracranial hypertension. When evaluating headaches, clinicians should be aware of ophthalmic signs and symptoms that can narrow the differential diagnosis. On the other hand, when clinicians suspect a non-benign etiology, awareness of potential vision-threatening ophthalmologic manifestations can facilitate their prompt identification and treatment should they be present. Since many non-benign headaches involve a rise in ICP, recognition of symptoms typical of intracranial hypertension is paramount to facilitate appropriate workup. Red flag symptoms such as positional worsening of headache, nausea, vomiting, pulsatile tinnitus, diplopia, new-onset esotropia, persistent sun-setting eyes, and/or TVOs warrant fundoscopic exam to evaluate for optic disc edema and urgent neuroimaging (especially if disc edema is noted).

## Data Availability

Not applicable.

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
