# Peer review of "When Pediatric Headaches Are Not Benign—Eye Findings"

_children, 2023, doi:10.3390/children10020372_

Round 1
Reviewer 1 Report
This is a review article, written for an audience of pediatricians, discussing life and/or vision-threatening etiologies of headache in children. This is an important topic, relevant to this audience.
I have a some suggestions that may improve the review:
1. In Section 3.2, it may be helpful to briefly clarify the relationship of ADEM with MOG-IgG associated disease (MOGAD) in children, since there is quite a bit of overlap (ie, MOGAD is the underlying cause for many cases of ADEM)
2. Also in Section 3.2, when discussing RAPD as a typical sign of optic neuritis, it may be worth noting that when optic neuritis is bilateral (as can be the case in ADEM), there may not be any RAPD
3. Section 3.3, Line 222: correct spelling of Rocky Mountain spotted fever
4. Section 3.3, Line 240: AIS acronym used, but this was never defined
5. Section 3.5.1
a. The description of “convergence retraction nystagmus” is a bit inaccurate. In fact, it is not a true nystagmus, it just represents co-contraction of the medial and lateral recti on attempted upgaze
b. Discussion/mention of pseudo-Argyll Robertson pupils is probably superfluous, and simply indicating the presence of light-near dissociation in a dorsal midbrain syndrome would clarify the finding
6. Section 3.6.3
a. Venous sinus stenosis only resolves with normalization of ICP in a minority of cases and, in fact, it has become a key treatment target for IIH (ie, with venous sinus stenting). For line 366, consider adding “in some cases” to the end of the sentence to clarify this.
b. Calling something “secondary IIH” does not really make sense- how can it be secondary to a known factor and idiopathic? Please clarify terminology.
7. Section 3.6.4
a. Presence of cranial neuropathies aside from CN VI actually excludes a diagnosis of IIH according to the modified Dandy criteria.
8. Section 3.6.5
a. Evaluation of CSF for cytology requires a large volume tap and is definitely not standard of care for suspected IIH
b. Venous sinus stenosis should be added to the characteristic imaging findings in the setting of high ICP
9. Section 3.6.6, conservative management often involves medication but not always. It can also involve observation (in mild cases) and weight loss if patient is overweight/obese
10. Overall: A short Conclusion summary highlighting the red flag symptoms and signs that should prompt emergent/urgent neuroimaging in a child presenting with headache may make emphasize the take-home points for the pediatrician.
Author Response
- In Section 3.2, it may be helpful to briefly clarify the relationship of ADEM with MOG-IgG associated disease (MOGAD) in children, since there is quite a bit of overlap (ie, MOGAD is the underlying cause for many cases of ADEM
Response: We have added this clarification in the manuscript.
- Also in Section 3.2, when discussing RAPD as a typical sign of optic neuritis, it may be worth noting that when optic neuritis is bilateral (as can be the case in ADEM), there may not be any RAPD.
Response: We have made this addition to the manuscript.
- Section 3.3, Line 222: correct spelling of Rocky Mountain spotted fever
Response: We have made these correction in the manuscript.
- Section 3.3, Line 240: AIS acronym used, but this was never defined
Response: We have added the definition of this acronym in the manuscript.
- Section 3.5.1
- The description of “convergence retraction nystagmus” is a bit inaccurate. In fact, it is not a true nystagmus, it just represents co-contraction of the medial and lateral recti on attempted upgaze
Response: We have made these changes in the manuscript.
- Discussion/mention of pseudo-Argyll Robertson pupils is probably superfluous, and simply indicating the presence of light-near dissociation in a dorsal midbrain syndrome would clarify the finding
Response: We have made these changes in the manuscript.
- Section 3.6.3
- Venous sinus stenosis only resolves with normalization of ICP in a minority of cases and, in fact, it has become a key treatment target for IIH (ie, with venous sinus stenting). For line 366, consider adding “in some cases” to the end of the sentence to clarify this.
Response: We have made this addition to the manuscript.
- Calling something “secondary IIH” does not really make sense- how can it be secondary to a known factor and idiopathic? Please clarify terminology.
Response: This is very reasonable. We went ahead and removed the term ‘secondary IIH’ and instead describe factors associated with development of IIH without using this contradictory terminology.
- Section 3.6.4
- Presence of cranial neuropathies aside from CN VI actually excludes a diagnosis of IIH according to the modified Dandy criteria
Response: We have removed discussion of cranial nerve involvement other than CN VI.
- Section 3.6.5
- Evaluation of CSF for cytology requires a large volume tap and is definitely notstandard of care for suspected IIH
Response: We have removed our statement regarding CSF cytology as part of work up for suspected IIH from the manuscript.
- Venous sinus stenosis should be added to the characteristic imaging findings in the setting of high ICP
Response: We have made this addition to the manuscript.
- Section 3.6.6, conservative management often involves medication but not always. It can also involve observation (in mild cases) and weight loss if patient is overweight/obese
Response: We have made this clarification in the manuscript.
- Overall: A short Conclusion summary highlighting the red flag symptoms and signs that should prompt emergent/urgent neuroimaging in a child presenting with headache may make emphasize the take-home points for the pediatrician.
Response: We have made this addition to the manuscript.
Reviewer 2 Report
Well written overview, clearly states clincal relevant information regarding diagnosis and treatment
Author Response
Well written overview, clearly states clincal relevant information regarding diagnosis and treatment
Response: Thank you for your review. We have made changes according to Reviewer #1’s suggestions.